# Characterization of Buckwheat Beverages Fermented with Lactic Acid Bacterial Cultures and Bifidobacteria

**DOI:** 10.3390/foods9121771

**Published:** 2020-11-29

**Authors:** Ewa Kowalska, Małgorzata Ziarno

**Affiliations:** 1Institute of Horticultural Sciences, Warsaw University of Life Sciences—SGGW (WULS-SGGW), 02-787 Warsaw, Poland; 2Department of Food Technology and Assessment, Division of Milk Technology, Institute of Food Science, Warsaw University of Life Sciences—SGGW (WULS-SGGW), 02-787 Warsaw, Poland; malgorzata_ziarno@sggw.edu.pl

**Keywords:** buckwheat, health, lactic acid bacteria, lactic acid fermentation, bifidobacterial, probiotics

## Abstract

This study aimed to examine the effect of four different industrial starter cultures containing lactic acid bacteria (LAB) and bifidobacteria on selected characteristics of beverages prepared from buckwheat and stored at 4 °C for 28 days. This study included the determination of pH during fermentation and during refrigerated storage, determination of the number of LAB and bifidobacteria, and chromatographic analysis of carbohydrates. This study showed that the tested starter cultures effectively fermented the buckwheat beverage. There was a sufficient number of viable cells in the starter microflora for the obtained beverages to exhibit potential health-promoting properties. Beverages had stable pH values during refrigerated storage. The stored beverages showed changes in the content of selected carbohydrates, which indicates the constant biochemical activity of the present starter microflora. This study provides useful references on the metabolism of LAB in plant-based beverages.

## 1. Introduction

Fermented plant-based products are a good substitute to dairy products, which cannot be consumed by people with food allergies or intolerance. Buckwheat, as a pseudocereal with a rich composition and high nutritional value, can be an ideal base for such products having a high proportion of lactic acid bacteria (LAB). Due to the fermentation process, they gain a pro-health value, while they are minimally processed. Furthermore, the probiotic LAB have a positive effect on human health by regulating the functions of the intestinal microbiota. They are used to prevent infections of the digestive system and increase immunity. They have anticancerogenic and antiallergenic effects [1]. Food intolerances are not related to the immune system and are caused by sensitivity to certain food ingredients, e.g., gluten [2]. Currently, an increasing number of people are diagnosed with this ailment, and almost 20% of the population has been affected by food intolerance [3]. The mechanisms of food intolerance are not known completely; however, they may be related to the neuroendocrine system of the digestive system [4]. Gluten intolerance can be possibly treated by excluding gluten from the diet. Any food containing gluten added in amounts harmful to patients should be avoided. According to the sources, harmful amounts are those exceeding 50–100 mg of gluten per day [5].

Nowadays, buckwheat is most abundant in China, Russia, Ukraine, and the USA. In Poland, the cultivation of buckwheat is limited by low temperatures; however, despite the production of this grain being much smaller than that in other countries, it still remains the most cultivated by its producers. Buckwheat is a dicotyledonous plant known as a pseudocereal and classified under the secondary plant group [6]. Buckwheat kernels are composed of gluten-free protein and balanced amino acid content [7]. Flour obtained from this grain is a rich source of minerals, such as copper, zinc, manganese, potassium, magnesium, phosphorus, and potassium [8]. It is high in polyphenols, including flavonoids such as rutin, orientin, vitexin, quercetin, isovitexin, and isoorientin [9]. Buckwheat achenes are also a source of many vitamins, including thiamine (3.3 mg/kg), riboflavin (10.6 mg/kg), niacin (18.0 mg/kg), pantothenic acid (11.0 mg/kg), and pyridoxine (1.5 mg/kg) [10]. 

Fermented products have been included in the human diet since the beginning of human civilization and have been the most fundamental food, and although people were not aware of it at the time, these products had a positive effect on their health [11]. They are defined as food obtained by the spontaneous or controlled growth of microorganisms and the enzymatic conversion of their main ingredients. Currently, there is rapid development of this type of food, enabling the production of thousands of various products [12]. As consumers’ awareness of health and proper diet increases, interest in natural health products is also increasing—thus, there has been a “big comeback” for fermented foods. Today, natural products are promoted on the internet, press, and television, and the wide-ranging health effects attributed to these products have increased scientists’ interest in creating new products or improving the existing ones. Moreover, it has been proposed that fermented foods should be included in the dietary recommendations [13]. The fermentation process of certain food products leads them to gain new health properties and features that the original product did not have. A big advantage of this is the presence of beneficial microorganisms and the ability to prevent the development of pathogenic microflora [14]. 

In Asia and Africa, fermented plant beverages are mostly traditional products. In addition to their unusual, exceptional taste, they exhibit many healing properties and are used in the prevention of many diseases. An example of such a beverage is boza, which is popular in Turkey and Bulgaria. It is produced by fermenting a mixture of many cereals, including wheat, rye, rice, and corn [15]. A popular beverage in Africa is togwa, which is composed of millet, sorghum, and maize flour [16]. A similar beverage is mahewu. In this case, natural fermentation occurs, and malt flour is added to the aforementioned ingredients [17]. In addition to cereals, fruit or fruit juices are increasingly used for the production of fermented plant beverages [18].

The present study aimed to understand the effect of four different industrial bacterial starter cultures containing LAB and bifidobacteria on three features of buckwheat beverage: pH, bacterial viability, and sugar content.

## 2. Materials and Methods 

### 2.1. Buckwheat Grains

Unroasted buckwheat groats obtained from Agro Bio Test (Warsaw, Poland) were used in the study. The established nutritional value of buckwheat was as follows: energy value: 1659 kJ/392 kcal in 100 g; fat: 2.7 g in 100 g (including saturated fatty acids: 0.6 g in 100 g); carbohydrates: 75 g in 100 g (including sugars: 2.6 g in 100 g); protein: 12 g in 100 g; salt: 0.03 g in 100 g. Based on this information and on the prepared recipe of buckwheat beverages, their nutritional information was determined: fat, carbohydrate, and protein content.

### 2.2. Lactic Acid Bacterial and Bifidobacteria Strains

Four industrial bacterial starter cultures were used in this study: (a) ABY-3 (Chr. Hansen, Hoersholm, Denmark), thermophilic yoghurt culture containing: *Streptococcus thermophilus* and *Lactobacillus delbrueckii* subsp. *bulgaricus*, *Lactobacillus acidophilus* La-5, and *Bifidobacterium animalis* subsp. *lactis* BB-12; (b) YO-MIX 207 LYO (DuPont Danisco, Copenhagen, Denmark), thermophilic yoghurt culture containing: *S. thermophilus*, *L. delbrueckii* subsp. *bulgaricus*, *L. acidophilus*, and *B. lactis*; (c) YO-MIX 205 LYO (DuPont Danisco, Copenhagen, Denmark), thermophilic yoghurt culture containing: *S. thermophilus*, *L. delbrueckii* subsp. *bulgaricus*, *L. acidophilus*, and *B. lactis*; (d) VEGE 033 LYO (DuPont Danisco, Copenhagen, Denmark), thermophilic yoghurt culture containing: *S. thermophilus*, *L. delbrueckii* subsp. *bulgaricus*, *L. acidophilus* NCFM, and *B. lactis* HN019.

### 2.3. Preparation of Fermented Buckwheat Beverages

For the preparation, 200 g of raw buckwheat was weighed and boiled in 3000 mL of water for 25 min. This proportion led to the appropriate consistency of the beverage. Subsequently, the cooled groats and water were blended into a homogeneous mass and strained through a dense sieve to eliminate the groats. Finally, 150 mL of buckwheat mash was poured into jars and sterilized at 121 °C for 20 min.

Glass jars containing buckwheat beverages were maintained at 37 °C, and then an aliquot of the appropriate culture was added (0.06 g of starter per 1000 mL of beverage). Subsequently, the beverages were placed in an incubator at 37 °C for 5 h and stored at 6 °C for 28 days. 

### 2.4. Microbiological Analysis

The microbiological analysis was performed immediately before and after the fermentation process, as well as on the 7th and 28th day of storage of the beverage samples at 6 °C. Three repetitions were made of the determination of the number of each type of bacteria. Three microbiological media were used to conduct the analyses: MRS (de Man, Rogosa and Sharpe, a selective medium for the isolation of lactic acid bacteria ) agar (Merck, Darmstadt, Germany) to determine the number of lactobacilli cells, BSM agar (Bifidus Selective Medium Agar, Sigma-Aldrich, St. Louis, MO, USA) to determine the number of bifidobacteria cells, and M17 agar (Merck, Darmstadt, Germany) to determine the number of *S. thermophilus* cells. Microbiological analysis was performed by the traditional plate method. All media were incubated at 37 °C for 72 h. The Petri dishes with the MRS agar and BSM agar were inserted in the anaerobic jars. The results thus obtained are expressed as mean value as colony-forming unit per 1 mL of beverage (CFU/mL) in two parallel replicates.

### 2.5. Determination of the pH Value

The pH was measured using a pH-meter model CPO-505 (Elmetron, Zabrze, Poland). The determinations were made before the fermentation of the buckwheat beverage, during fermentation (after every full hour), and during the refrigerated storage period (after 7, 21, and 28 days). Three repetitions of pH measurements were made for each type of buckwheat beverage. The measurements were performed similarly for each beverage. 

### 2.6. Preparation of Extracts for Carbohydrate Analysis

First, carbohydrate extraction was performed. For this purpose, 8.0 g of the sample and 32.0 g of methanol (HPLC grade, > 0.9%, Sigma-Aldrich, St. Louis, MO, USA) were measured into a falcon tube, and the contents of the tube were mixed intensively and placed in an ultrasonic bath for 30 min (at 30 °C). The product thus obtained was then centrifuged in a laboratory centrifuge (MPW-350R, Irmeco, Bielsko-Biała, Poland) at 5200 rpm at 4 °C for 30 min. The obtained supernatant was transferred to another falcon tube and methanol was evaporated from the transferred supernatant by leaving the open falcon tubes in a fume cupboard in a water bath at 80 °C. The evaporation was continued until the volume reached approximately 4–5 mL. After evaporating the solvent and concentrating the samples, the contents of the test tubes were mixed intensively and filtered into 10 mL chromatographic vials through a syringe filter with a pore size of 0.45 µm. Chromatographic vials with prepared samples were frozen until chromatographic analysis.

### 2.7. Chromatographic Analysis of Carbohydrates

The carbohydrates were subjected to chromatographic analysis on a HPLC (high-performance liquid chromatography) device consisting of: DeltaChrom Pump Injector (S6020 Needle Injection Valve, Sykam, Fürstenfeldbruck, Germany), DeltaChrom Temperature Control Unit (Sykam), refractive index detector (S3580 RI Detector, Sykam, Eresing, Germany), precolumn Guard Column Sugar-D (10 mm × 4.6 mm, 5 μm; Cosmosil, Nacalai Tesque, Kyoto, Japan), and column Sugar-D (250 mm × 4.6 mm, 5 μm; Cosmosil, Kyoto, Japan). The mobile phase contained acetonitrile (HPLC grade, > 99.9%, Sigma-Aldrich, St. Louis, MO, USA) and ultrapure distilled water in a weight ratio of 60:40. From the chromatographic vials, 40 µL of the solution was withdrawn using a laboratory microsyringe. HPLC device operating parameters were as follows: flow 1 mL/min, column temperature 30 °C; RI detector settings: range 10000 mV and sample rate 2 Hz. Each sample was analyzed for 25 min in duplicate. Carbohydrates were identified, and their concentration in the tested samples was calculated based on a comparison of the obtained results with the results obtained for external standards. Relevant external standards of xylose, fructose, arabinose, glucose, melibiose, sucrose, and maltose (Sigma-Aldrich, St. Louis, MO, USA) were determined by analyzing the samples.

### 2.8. Statistical Analysis of Results

The results were statistically analyzed in Microsoft Excel program, in which mean values and standard deviations were calculated, and the R program, in which one-way ANOVA and Tukey’s test were performed (for *α* = 0.05).

## 3. Results

### 3.1. Basic Chemical Composition of Buckwheat Beverages

The calculated nutritional value of 100 g of buckwheat beverage was:Fat 0.16 g (including saturated acids: 0.04 g);Carbohydrates 4.69 g (including sugars: 0.16 g);Protein 0.75 g.

The average water content was 87.9%. The aforementioned calculations should be considered with some reservation (especially the calculated carbohydrate content), because the buckwheat used in this study was subjected to several processes that could affect the chemical composition, such as cooking, blending, and sterilization.

### 3.2. pH Values during Fermentation

Before the fermentation process, the pH of buckwheat beverage samples intended for fermentation with YO-MIX 207, YO-MIX 205, and ABY-3 was on average 6.550 and 6.400 in the case of the samples intended for fermentation with VEGE 033 (Figure 1). The fermentation of the beverage with YO-MIX 207 was the most effective followed by fermentation of the beverage with YO-MIX 205 culture. Within 1–2 h of fermentation, both the beverages reached an average pH value of 4.8, which was statistically significantly different from the pH value measured before the fermentation process (*p*-value < 0.05). ABY3 and VEGE 033 were less efficient in terms of acidification, wherein this value was not recorded until 3 and 4 h of the fermentation process. After fermentation for 5 h, all of the beverages reached a pH of 4.5–4.9.

### 3.3. Bacterial Population during Storage

After the fermentation of the beverages with YO-MIX 205 (Figure 2), the number of cells of all three types of bacteria decreased, in the case of bifidobacteria (from 7.8 log(CFU/mL) to 7.5 log(CFU/mL)) and in the case of lactobacilli (from 7.9 log(CFU/mL) to 7.6 log(CFU/mL)). The number of streptococcal cells decreased from 8.4 log(CFU/mL) to 7.8 log(CFU/mL). However, after 7 days of storage, the number of all bacteria increased again. On the 28th day of refrigerated storage, the buckwheat beverage samples contained an average of 7.6 log(CFU/mL) of live lactobacilli cells, 7.8 log(CFU/mL) of viable lactic streptococcal cells, and 7.5 log(CFU/mL) of viable bifidobacterial cells. Statistical analysis showed no significant differences in the number of LAB and bifidobacteria in the beverages stored under refrigerated conditions. The number of viable cells of lactobacilli, lactic streptococci, and bifidobacteria on the final day of the experiment was over 7 log(CFU/mL), which indicated the potential health-promoting properties. 

Similar results were obtained when determining the bacteria in beverages fermented with YO MIX 207. The lactobacilli cells showed a decrease in growth after fermentation (from 8.7 log(CFU/mL) to 7.8 log(CFU/mL)) and then an increase from 7.8 log(CFU/mL) to 8.0 log(CFU/mL) after 7 days of refrigeration. There were very similar changes in the number of streptococcal and bifidobacterial cells. On the 28th day of refrigerated storage, the buckwheat beverage samples contained an average of 7.8 log(CFU/mL) of live lactobacilli cells, 7.7 log(CFU/mL) of viable lactic streptococcal cells, and 7.7 log(CFU/mL) of viable bifidobacterial cells. during refrigerated storage of the beverage fermented with YO-MIX 207 culture, there were statistically significant changes in the number of lactobacilli, similarly to bifidobacteria. Statistical analysis showed no significant differences in the number of lactic streptococcal cells.

As for the fermented beverages with the other discussed starter cultures, after 7 days of refrigerated storage, a slight change was observed in the number of lactobacilli, lactic streptococci, and bifidobacteria; however, it was not statistically significant. After 28 days of storage of beverage samples at 4 °C, the average bacterial cell count was 8.0 log(CFU/mL) for lactobacilli, 7.8 log(CFU/mL) for lactic streptococci, and 8.0 log(CFU/mL) for bifidobacteria.

The smallest fluctuations in the number of bacterial cells occurred in beverages fermented with ABY-3 culture. Contrary to buckwheat beverages fermented with YO-MIX 207 or YO-MIX 205 bacterial cultures, there were no such significant changes in the number of bacterial cell populations. After fermentation, the number of bifidobacterial cells decreased the most (statistically significant). There were no significant changes in the number of lactobacilli and lactic streptococci cells during the refrigerated storage of samples of buckwheat beverages fermented with ABY-3 culture. After 28 days of storage of the beverage samples at 4 °C, the average bacterial cell count was 8.0 log(CFU/mL) for lactobacilli, 7.8 log(CFU/mL) for lactic streptococci, and 8.0 log(CFU/mL) for bifidobacteria. The number of viable cells of lactobacilli, lactic streptococci, and bifidobacteria on the final day of the experiment was over 7 log(CFU/mL).

In buckwheat beverages fermented with VEGE 033 (Figure 2), lactic streptococci constituted the largest proportion of the prefermentation bacterial cell population, which, after the fermentation process, was statistically significantly decreased. A statistically significant change in the number of bifidobacterial cells occurred only on the 7th day of storage of the samples—8.2 log(CFU/mL). VEGE 033 showed the lowest number of bifidobacterial cells (during the entire cold storage period). As for the remaining studied groups of bacteria, in comparison to beverages fermented with other cultures, the number of streptococcal cells on the 7th day of storage was recorded to be on average 8.2 log(CFU/mL) in the samples of buckwheat beverages fermented with VEGE 033. On the 28th day of refrigerated storage, similar numbers of live cells of all types of bacteria were found: on average 7.7 log(CFU/mL) of lactobacilli, 7.5 log(CFU/mL) of lactic streptococci, and 7.4 log(CFU/mL) of bifidobacteria.

### 3.4. pH Values during Refrigerated Storage

The most stable pH value during refrigerated storage was exhibited by beverages fermented with VEGE 033, ABY-3, or YO-MIX 205 (Figure 3). Changes in the pH value during 28 days of refrigerated storage were noted in the beverage samples fermented with YO-MIX 207. The results obtained after measuring the pH of buckwheat beverages fermented with YO-MIX 205, ABY-3, and VEGE 033 cultures did not differ statistically. As can be seen from the data presented in Figure 3, these pH changes depend on the starter culture used; therefore, it can be assumed that they depend on the qualitative and quantitative composition of the microflora present during fermentation, and not on the composition and properties of the fermented buckwheat matrix.

### 3.5. Carbohydrates Content

The examined buckwheat beverages contained seven types of sugars: xylose, fructose, arabinose, glucose, melibiose, sucrose, and maltose. 

Significant changes were observed in the total carbohydrate content in the beverages (Table 1). The initial (before fermentation) total content of all tested carbohydrates in the fermented buckwheat beverage was 4.598 g in 100 g of the product. As a result of the processes performed on buckwheat beverages, primarily cooking and sterilization in an aqueous solution, certain complex carbohydrates and polysaccharides were decomposed, which released some of them, i.e., xylose, fructose, arabinose, glucose, melibiose, sucrose and maltose, which were determined by chromatography. After the fermentation of the buckwheat beverage samples, the highest total carbohydrate content was found in the beverages fermented with ABY-3 culture, while the lowest content was found in the beverages fermented with culture YO-MIX 207. For example, after the first week of refrigerated storage, the highest total content of all tested carbohydrates was measured in the beverage fermented with the VEGE 033 culture and the lowest in the beverage fermented with YO-MIX 207 culture. In the case of beverages fermented with the VEGE 033 or YO-MIX 205 culture, the total content of all tested carbohydrates on the seventh day of storage was the highest compared to that immediately after fermentation. In beverages fermented with ABY-3 and YO MIX 207 cultures, the total content of all tested carbohydrates decreased. After an additional 21 days of storage, in all the beverages trialed, a significant reduction in the tested total carbohydrate content was measured. There was a slight change in the carbohydrate content in the beverage fermented with YO-MIX 207 culture and the largest change in the beverage fermented with the VEGE 033 culture.

## 4. Discussion

Our study was the first to use buckwheat as a plant matrix. We compared the results of our experiment with the results of studies conducted with the use of other plant matrices, but of the same bacterial strains (though not all). Our study was the first to use four mixed bacterial cultures.

The number of live LAB is the most important feature, as it indicates the quality of the probiotic beverages. The number of microorganism cells at 7–8 log(CFU/mL) indicates that a product has probiotic properties [19]. In this study, such numbers of live LAB and bifidobacteria were obtained; however, lower values were obtained in studies on rice beverages. Before fermentation, the number of bacterial cells was lower than that in buckwheat beverages, and the population of lactic bacteria was 5.0 log(CFU/mL). However, after the 16 h fermentation process, the number of bacterial cells increased to 8.1 log(CFU/mL) and remained constant until the completion of the fermentation process [20]. Similar results were obtained after the fermentation of corn or rice-based beverages, in which the microbial cell population was at the level of 7–8 log(CFU/mL) [21].

There were some similarities between the results obtained in this study and in the study that used different strains of LAB for the fermentation of soy milk, including *L. delbrueckii* subsp. *bulgaricus* and *L. acidophilus*, which were also used in this work. The cell population of all cultures was 8 log(CFU/mL), and a similar concentration of bacterial cells was obtained in buckwheat beverages. In each bacterial culture, *L. delbrueckii* subsp. *bulgaricus* and *L. acidophilus* were present but differed at the strain level [22].

In this study, in all the products based on plant substrates, similarities were noted in the results of the lactic bacterial population, despite different *Lactobacillus* strains being used. The effective growth of LAB in plant-based beverages can be explained by the presence of high amounts of mono- and disaccharides in the plant media. On the other hand, according to another study, the observed numbers of bacterial cells after the cold storage period may result from the production of antimicrobial compounds by bacteria (e.g., hydrogen peroxide, bacteriocins, or organic acids) [23].

A similar frequency of changes in the number of LAB was noted in the study on bean beverages. The final bacterial count after 28 days of refrigerated storage was 6.9 log(CFU/mL) [24]. Upon comparison, a slightly higher number of viable lactobacilli cells was found in the buckwheat beverages. Better bacterial growth on buckwheat substrate could have been due to higher content and availability of sugar.

In this study, the obtained pH values indicated an effective 5 h fermentation process by LAB. The fermentation process by each of the industrial cultures stabilized the final pH below 5. Similar pH values were obtained after the fermentation of soybean beverage by *S. thermophilus*, at 4.65 [25]. The pH value was greater than that in barley malt fermented with *Lactobacillus plantarum* (NCIMB 8826) and *L. acidophilus* (NCIMB 8821) strains at 30 °C recorded at approximately 4.0 [26]. The differences between the cited results and those obtained in this study may be due to the specificity of plant matrices, as well as the use of various bacterial cultures in the research. 

Unfortunately, no results confirming the effective fermentation process during the whole refrigerated storage of buckwheat beverages were obtained. Studies conducted on bean beverages with industrial cultures ABY-3 and YO-MIX yielded results similar to those obtained in this study, but only until the 21st day of refrigerated storage. The pH values remained at the levels of 4.34 and 4.29. After 28 days of refrigerated storage, these values slightly decreased to 4.34 and 4.27, which indicated an active fermentation process [24].

It can be assumed that the changes in the tested carbohydrate content during 28 days of refrigerated storage of fermented buckwheat beverages were due to many changes occurring in the analyzed samples: biochemical activity of LAB and bifidobacteria, as well as enzymatic changes of the present polysaccharides. It is difficult to compare the results of this work with the results of other researchers, due to the lack of publications in which similar measurements have been made. The results differ from those obtained in this study during examination of the sugar content in the cooked buckwheat wort. In the tested samples, glucose constituted the highest concentration, while there were traces of other potentially fermentable sugars [27]. On the other hand, in this study, sucrose was the carbohydrate occurring in the highest quantity after fermentation and at subsequent storage stages of the samples fermented with YO-MIX 207, YOMIX 205, or ABY-3 cultures, most likely due to starch decomposition.

Other researchers have described the profile of sugars found in buckwheat, where sucrose constituted the highest concentration, while carbohydrates such as xylose, glucose, arabinose, and melibiose were present in much smaller amounts [28]. Another study has reported that with an increasing amount of water and prolonged heating time, the glucose content increased [29]. Our work is distinguished by an experiment to check exactly what sugars are present in a buckwheat beverage and what is their exact content. Knowledge about changes in sugar content at the individual stages of buckwheat beverage storage gave information about the biochemical activity of lactic acid bacteria and bifidobacteria, which allowed us to draw conclusions about their viability in the beverage.

## 5. Conclusions

The results obtained in this study show that buckwheat is a well-fermentable medium; therefore, it can be an alternative to fermented milk beverages. Gluten-free cereal beverages are a response to the growing demand for such products among people suffering from celiac disease and food intolerances. High bacterial survival during the storage period enables achieves a therapeutic effect similar to that caused by fermented milk products, such as kefir, buttermilk, or yoghurt. An additional advantage of the product is the lack of allergenic milk proteins. Due to the consumption of milk and other dairy products, an increasing number of people are experiencing side effects such as gas, indigestion, and diarrhea, which leads to their exclusion from their diet. In such a case, dietary supplements containing probiotic strains are often used to supplement the intestinal microflora and increase the body’s immunity. The fermented buckwheat beverages can replace these types of supplements and provide other essential nutrients for the body. The product is catered not only to people suffering from digestive system dysfunctions; healthy people who follow a balanced diet and a healthy lifestyle can also benefit from consuming fermented buckwheat beverages. In buckwheat beverages fermented with starter cultures and stored under refrigeration for 28 days, changes in the content of selected carbohydrates are observed, which proves the constant biochemical activity of the present starter microbial flora: lactic bacteria and bifidobacteria. In addition to LAB and bifidobacteria, the base of the buckwheat beverage is important, as it was a medium required for the growth of the bacterial population used for fermentation. Considering the results obtained from individual measurements and analyses, buckwheat is a pseudocereal that is successfully fermented by LAB and bifidobacteria.

## Figures and Tables

**Figure 1 foods-09-01771-f001:**
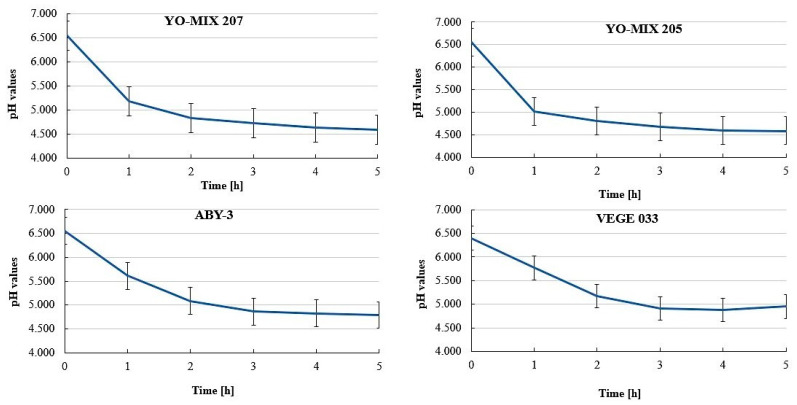
Acidification curves of buckwheat beverages during the fermentation process (mean values and standard deviations, *n* = 3).

**Figure 2 foods-09-01771-f002:**
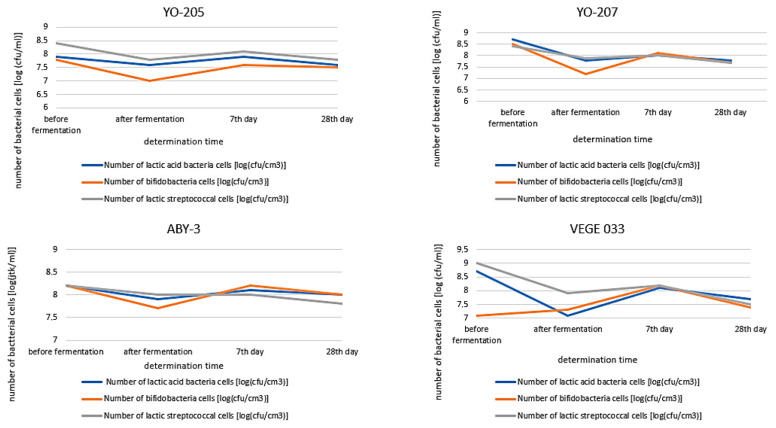
Population of lactic acid bacteria and bifidobacteria in buckwheat beverages fermented by starter cultures (mean values and standard deviations, *n* = 3).

**Figure 3 foods-09-01771-f003:**
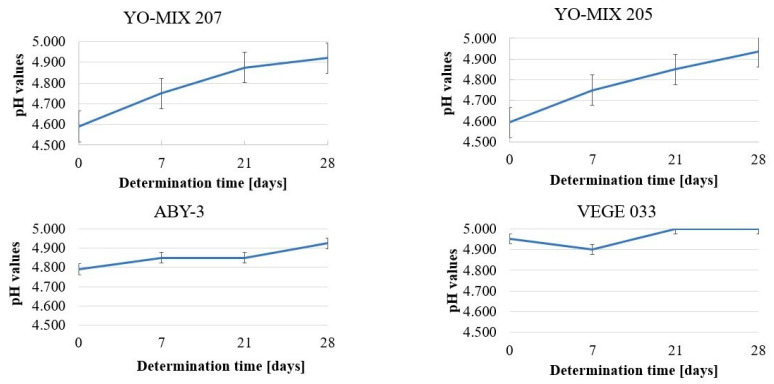
pH values of buckwheat beverages during refrigerated storage (mean values and standard deviations, *n* = 3).

**Table 1 foods-09-01771-t001:** Content of carbohydrates in buckwheat beverages fermented by starter cultures (mean values and standard deviations, *n* = 2).

	Carbohydrates (g/100 g beverage)	Before fermentation	After fermentation	After 7 days of storage	After 28 days of storage
**YO-MIX 205**	Xylose	0.00 ± 0.00	0.00 ± 0.00	0.04 ± 0.00	0.23 ± 0.01
	Fructose	0.10 ± 0.00	0.10 ± 0.00	0.09 ± 0.00	0.24 ± 0.01
	Arabinose	0.00 ± 0.00	0.15 ± 0.01	0.87 ± 0.04	0.00 ± 0.00
	Glucose	2.96 ± 0.15	0.00 ± 0.00	0.00 ± 0.00	0.12 ± 0.01
	Melibiose	0.00 ± 0.00	0.29 ± 0.01	0.19 ± 0.01	0.06 ± 0.00
	Sucrose	1.54 ± 0.08	1.51 ± 0.08	1.76 ± 0.09	0.60 ± 0.03
	Maltose	0.22 ± 0.01	0.00 ± 0.00	0.00 ± 0.00	0.00 ± 0.00
	All	4.60 ± 0.23	2.05 ± 0.10	2.96 ± 0.15	1.27 ± 0.06
**YO-MIX 207**	Xylose	0.00 ± 0.00	0.00 ± 0.00	0.00 ± 0.00	0.14 ± 0.01
	Fructose	0.10 ± 0.00	0.12 ± 0.01	0.08 ± 0.00	0.16 ± 0.01
	Arabinose	0.00 ± 0.00	0.07 ± 0.00	0.09 ± 0.00	0.00 ± 0.00
	Glucose	2.96 ± 0.15	0.20 ± 0.01	0.19 ± 0.01	0.10 ± 0.01
	Melibiose	0.00 ± 0.00	0.00 ± 0.00	0.00 ± 0.00	0.09 ± 0.00
	Sucrose	1.54 ± 0.08	1.44 ± 0.07	1.39 ± 0.07	0.75 ± 0.04
	Maltose	0.22 ± 0.01	0.00 ± 0.00	0.00 ± 0.00	0.00 ± 0.00
	All	4.60 ± 0.23	1.82 ± 0.09	1.74 ± 0.09	1.25 ± 0.06
**ABY-3**	Xylose	0.00 ± 0.00	0.13 ± 0.01	0.06 ± 0.00	0.19 ± 0.01
	Fructose	0.10 ± 0.00	0.32 ± 0.02	0.13 ± 0.01	0.00 ± 0.00
	Arabinose	0.00 ± 0.00	0.24 ± 0.01	0.29 ± 0.01	0.00 ± 0.00
	Glucose	2.96 ± 0.15	0.28 ± 0.01	0.25 ± 0.01	0.15 ± 0.01
	Melibiose	0.00 ± 0.00	0.00 ± 0.00	0.32 ± 0.02	0.15 ± 0.01
	Sucrose	1.54 ± 0.08	2.30 ± 0.12	1.59 ± 0.08	0.70 ± 0.03
	Maltose	0.22 ± 0.01	0.00 ± 0.00	0.00 ± 0.00	0.00 ± 0.00
	All	4.60 ± 0.23	3.27 ± 0.16	2.65 ± 0.13	1.20 ± 0.06
**VEGE 033**	Xylose	0.00 ± 0.00	0.08 ± 0.00	0.63 ± 0.03	0.71 ± 0.04
	Fructose	0.10 ± 0.00	0.09 ± 0.00	0.00 ± 0.00	0.00 ± 0.00
	Arabinose	0.00 ± 0.00	0.74 ± 0.04	1.30 ± 0.06	0.26 ± 0.01
	Glucose	2.96 ± 0.15	0.24 ± 0.01	0.33 ± 0.02	0.11 ± 0.01
	Melibiose	0.00 ± 0.00	0.00 ± 0.00	0.00 ± 0.00	0.00 ± 0.00
	Sucrose	1.54 ± 0.08	1.24 ± 0.06	1.60 ± 0.08	0.34 ± 0.02
	Maltose	0.22 ± 0.01	0.00 ± 0.00	0.00 ± 0.00	0.00 ± 0.00
	All	4.60 ± 0.23	2.39 ± 0.12	3.85 ± 0.19	1.41 ± 0.07

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
