# Peer review of "Characterization of Buckwheat Beverages Fermented with Lactic Acid Bacterial Cultures and Bifidobacteria"

_foods, 2020, doi:10.3390/foods9121771_

Round 1

Reviewer 1 Report

The manuscript entitled “The possibility of obtaining buckwheat beverages fermented with lactic acid bacterial cultures and bifidobacteria”  is presented well and need some changes and explanation before for consideration for publication

Following are my comments and suggestions:

Line: 89-96: Please write the scientific names in italics

Figure 1: very poor resolution, please fix this problem

Figure 2: very poor resolution, please fix it

The legends of the figure needs to be explained in details, the present legend is not acceptable

Since the pH has fallen down drastically within 5 hrs, Can you please describe in details, how the starter culture was prepared: was it powder ? was it pellet  or broth ?, please correct accordingly

Line 271 – 272: write scientific names in italics

Such error is available in several places, please check it properly

Figure 3: the increment of pH value is shown ? please explain with valid references ?

How can we know the proportion of two lactic cultures counted in MRS agar ?, Do you have data on different count between two lactobacillus strains ?

Please correct/address all the comments, modify your manuscript according and resubmit it.

Reviewer 2 Report

Ms. ID Foods 992864: The possibility of obtaining buckwheat beverages fermented with lactic acid bacterial cultures and bifidobacteria

General comments:

This study is mainly based on the comparison of four commercial starters in order to obtain a buckwheat beverage. The aim is to obtain non-dairy products.  The subject is therefore not original but the substrate (buckwheat) has been little studied to produce fermented beverages.

Major concerns:

  1. Results based on pH and viability values for both production (Figure 1 pH, Figure 2 viable) and storage (Figure 3 pH) are contradictory. How the authors can explain that the pre- and post-fermentation counts remain stable or decrease and that the pH decreases. The production of organic acids (lactic and acetic acids) occurs after growth. It is possible that the very high level of inoculation may explain the decrease in pH. Measuring the titratable acidity developed would have made it possible to know this. During storage, pH values increase (fig. 3) which is contradictory to what is observed with products such as yogurt. The presence of lactobacilli often results in post-acidification. Hydrolysis of sugars does not support an increase in pH unless proteolysis has taken place. The bacteria used are not considered very proteolytic. An increase in pH during storage may suggest microbial contamination. Has the final product been enumerated for contaminants and yeasts?
  2. The conclusions are not clearly stated and based on the results obtained.

Specific comments:

Title: The possibility of obtaining buckwheat beverages fermented with lactic acid bacterial cultures and bifidobacteria

And

Lines 317-318: The results obtained in this study show the high potential of fermented buckwheat beverage as a probiotic product with prohealth properties.

Comments:

The suggested title contradicts this statement. The survival of bacteria such as bifidobacteria at levels above 10e7 cfu/mL allows the product to be recognized as probiotic according to legislations in various countries. However, the purpose of this study is not to demonstrate that the resulting buckwheat beverages has healthy properties.

The title should be revised to take into account the following statement

Lines 331-332: Considering the results obtained from individual measurements and analyses, buckwheat is a pseudocereal that is successfully fermented by LAB and bifidobacteria.

Suggestion Title: Characterization of buckwheat beverages fermented with lactic acid bacterial cultures and bifidobacteria

Lines 77-78: The present study aimed to understand the effect of four different industrial bacterial starter cultures containing LAB and bifidobacteria on selected features of buckwheat beverage. Please specify which selected features  ( pH, viability, sugar content?)

Materials and methods:

  1. The number of replicates for experiments and analyses is not clearly indicated. The legends of the figures specify n=3 but it would be important to make it clear that these are replicates in the text.

Results

Table 1: Please specify the number of replicates in the legend. Please also add the standard deviation for each value. Please the number with only two digits after the decimal point for all values.

Conclusions

Lines 317-318: The results suggest that the survival of LAB and bifidobacteria during fermentation and storage was maintained at levels appropriate in buckwheat beverages to be considered as probiotic products. There is no evidence that this product has prohealth properties. Please modify

Lines 333-334: Their proven health properties mean that buckwheat beverages can be used to prevent lifestyle diseases such as diabetes, obesity, or cancer. Comments: Without a clinical study on the product, it is not possible to extrapolate the benefits of probiotics. This statement should be omitted or tempered.

Minor comments:

Lines 89-94 and throughout the text: Please italicize the name of microorganisms

Figure 2 X-axis, please replace cm3 by mL = log (CFU/mL)

Reviewer 3 Report

The authors have investigated the effect of different starter cultures on the selected characteristics of buckwheat beverages. The data are not sufficient to make this study scientifically sound. This study needs more experiments, for example, analysis of nutrients and antioxidant potentials. There are many things to improve in the manuscript. The scientific names are also not italicized.

Introduction: What is the research gap that this research intends to address? Why this study is useful? What are the findings of other similar studies? Is this the first study to investigate the effect of four industrial bacterial starter cultures?

Materials and Methods: Why did the authors measure only the pH and carbohydrate of the beverage? What about variation in other nutrients before and after the fermentation/refrigerated storage? Is the buckwheat beverage prepared only for the dietary carbohydrate supplements?

Conclusion: Is this the first study to show a successful fermentation of buckwheat using LAB and bifidobacteria?

Please refer to the attached file for specific comments and suggestions.

Round 2

Reviewer 2 Report

No additional comments

Reviewer 3 Report

The manuscript has been improved. I still suggest citing relevant reports in the Introduction section.
